# Enhanced Cerebroprotection of Xenon-Loaded Liposomes in Combination with rtPA Thrombolysis for Embolic Ischemic Stroke

**DOI:** 10.3390/biom13081256

**Published:** 2023-08-16

**Authors:** Tao Peng, Keith Booher, Melanie R. Moody, Xing Yin, Jaroslaw Aronowski, David D. McPherson, Sean I. Savitz, Hyunggun Kim, Shao-Ling Huang

**Affiliations:** 1Division of Cardiovascular Medicine, Department of Internal Medicine, McGovern Medical School, The University of Texas Health Science Center at Houston, Houston, TX 77030, USA; tao.peng@uth.tmc.edu (T.P.); melanie.r.moody@uth.tmc.edu (M.R.M.); xing.yin@uth.tmc.edu (X.Y.); david.d.mcpherson@uth.tmc.edu (D.D.M.); 2Zymo Research Corporation, Irvine, CA 92614, USA; kbooher@zymoresearch.com; 3Department of Neurology, McGovern Medical School, The University of Texas Health Science Center at Houston, Houston, TX 77030, USA; j.aronowski@uth.tmc.edu (J.A.); sean.i.savitz@uth.tmc.edu (S.I.S.); 4Institute for Stroke and Cerebrovascular Disease, McGovern Medical School, The University of Texas Health Science Center at Houston, Houston, TX 77030, USA; 5Department of Biomechatronic Engineering, Sungkyunkwan University, Suwon 16419, Republic of Korea

**Keywords:** xenon, liposome, tissue plasminogen activator, cerebroprotective, embolic ischemic stroke

## Abstract

Xenon (Xe) has shown great potential as a stroke treatment due to its exceptional ability to protect brain tissue without inducing side effects. We have previously developed Xe-loaded liposomes for the ultrasound-activated delivery of Xe into the cerebral region and demonstrated their therapeutic efficacy. At present, the sole FDA-approved thrombolytic agent for stroke treatment is recombinant tissue plasminogen activator (rtPA). In this study, we aimed to investigate the potential of combining Xe-liposomes with an intravenous rtPA treatment in a clinically relevant embolic rat stroke model. We evaluated the combinational effect using an in vitro clot lysis model and an in vivo embolic middle cerebral artery occlusion (eMCAO) rat model. The treatment groups received intravenous administration of Xe-liposomes (20 mg/kg) at 2 h post-stroke onset, followed by the administration of rtPA (10 mg/kg) at either 2 or 4 h after the onset. Three days after the stroke, behavioral tests were conducted, and brain sections were collected for triphenyltetrazolium chloride (TTC) and TUNEL staining. Infarct size was determined as normalized infarct volume (%). Both in vitro and in vivo clot lysis experiments demonstrated that Xe-liposomes in combination with rtPA resulted in effective clot lysis comparable to the treatment with free rtPA alone. Animals treated with Xe-liposomes in combination with rtPA showed reduced TUNEL-positive cells and demonstrated improved neurological recovery. Importantly, Xe-liposomes in combination with late rtPA treatment reduced rtPA-induced hemorrhage, attributing to the reduction of MMP9 immunoreactivity. This study demonstrates that the combined therapy of Xe-liposomes and rtPA provides enhanced therapeutic efficacy, leading to decreased neuronal cell death and a potential to mitigate hemorrhagic side effects associated with late rtPA treatment.

## 1. Introduction

Stroke remains the second leading cause of mortality and disability on a global scale, resulting in approximately 11.6 million deaths and a loss of 116 million disability-adjusted life years in 2019 [1]. On average, this translates to an incidence of one stroke every three seconds. In the United States, stroke is a major cause of long-term disability and ranks as the fifth leading cause of death [2]. A stroke, regarded as a medical emergency, occurs when the normal blood flow to the brain is disrupted, leading to two main types: ischemic stroke, caused by blockage, and hemorrhagic stroke, caused by bleeding. Ischemic strokes account for approximately 87% of all stroke cases [3].

At present, recombinant tissue plasminogen activator (rtPA) is the only FDA-approved therapeutic intervention for ischemic stroke. Timely initiation of rtPA treatment is a critical aspect of stroke management and has demonstrated markedly improved outcomes for suitable patients. However, it is crucial to initiate rtPA treatment within a limited 4.5 h window from the onset of stroke symptoms. Beyond this time window, the potential for complications such as intracerebral hemorrhage significantly escalates. Furthermore, delayed rtPA treatment is correlated with an increased risk of hemorrhagic transformation and the deterioration of stroke outcomes [4].

Upon the occurrence of an ischemic stroke, interruption of the blood supply to a specific region of the brain instigates a series of events potentially leading to brain damage and disability. These events encompass excitotoxicity [5,6], oxidative stress, inflammation [7], and apoptosis (programmed cell death) [8]. Within the context of a stroke, the ischemic core signifies the central zone of the affected brain tissue, which undergoes irreversible damage due to blood flow deprivation. The area of brain tissue that surrounds the ischemic core remains viable but is vulnerable to damage if blood flow is not rapidly restored. The application of rtPA effectively dissolves the clot and reinstates blood flow to the affected area. Cytoprotection prior to rtPA treatment is aimed at preserving the integrity of the healthy brain tissue surrounding the core and potentially salvaging additional brain tissue.

Considering the complexity of the pathological cascade associated with stroke, an additional approach is to directly target the ischemic cascade to protect all cellular components of the neurovascular unit including neurons, astrocytes, microglia, pericytes, and endothelia within the ischemic region. This is known as cytoprotection [9]. Cytoprotective strategies have encountered various challenges. Many cytoprotective agents possess a limited capacity to cross the blood-brain barrier (BBB), which impedes their effective delivery to the brain. Furthermore, some cytoprotective therapies are not validated in conjunction with the existing rtPA treatment [10].

Xenon (Xe), a noble gas, offers promise as a potential cytoprotectant due to its capacity to readily diffuse into target tissues owing to its small size. Unlike many therapeutic agents, Xe can traverse the BBB and has exhibited its promising potential as an effective cytoprotective agent in stroke [10,11].

Xe provides immediate and enduring neuroprotection post brain injury. Distinct from other neuroprotective agents, Xe is an exceptional cytoprotective gas with pleiotropic effects. The potential mechanisms include stabilizing brain cell membranes by interacting with NMDA receptors and opening K+ATP channels, which in turn inhibit cytotoxicity and subsequent cell death. Our research, along with others’, has demonstrated that the protective influence of Xe on the brain extends beyond neurons, encompassing various elements of the neurovascular network [12,13,14]. Both proteomic and histologic analyses demonstrated that Xe inhibited MMP9 expression, which may stabilize BBB. These neurovascular cytoprotective effects stem from gene regulation [15] and Xe-based pre- or post-conditioning [16,17,18,19], initiating an array of innate protective pathways. This cascade involves the elevation of brain-derived neurotrophic factor (BDNF) expression [20]; the activation of pro-survival proteins such as Bcl-2 that enhance cellular tolerance to ischemic injury [16,21]; and the induction of hypoxia-inducible factor 1 alpha (HIF-1α), heme oxygenase 1 (HO-1) [22], and high mobility group protein B1 (HMGB-1) [22]. Remarkably, this cytoprotective conditioning induced by Xe has demonstrated a significant reduction in ischemic cerebral injury [16].

Inhalation is a conventional route for administering medical gases; the delivery of Xe through inhalation is rather complex, necessitating intubation, ventilation, and high Xe concentrations, which consequently reduces the maximum fraction of inspired oxygen. To overcome these challenges, we have leveraged our previous work on gas encapsulation within liposomes to design a highly efficient Xe delivery system to the brain, demonstrating the therapeutic efficacy of this approach in the treatment of stroke [12,13,23,24].

Given that rtPA remains the sole FDA-approved treatment for ischemic stroke, our study holds clinical significance as it aims to evaluate the therapeutic efficacy and compatibility of Xe-liposomes when used in conjunction with rtPA treatment in a preclinical animal model. A rat embolic stroke model, which closely mimics the pathophysiology and clinical features of human ischemic stroke, was selected for our investigation. In order to simulate embolic ischemic stroke, a blood clot was introduced into the middle cerebral artery (MCA), resulting in the occlusion of a cerebral blood vessel and the formation of an ischemic lesion. Reperfusion was achieved through the breakdown of the clot by administering rtPA. The extent of the brain tissue damage and the recovery of neurological behavior in ischemic stroke were examined following the treatment of Xe-liposomes in conjunction with both the early and late treatment of thrombolytic agents.

## 2. Materials and Methods

### 2.1. Preparation of Xe-Liposomes

Liposomes were composed of 1,2-dipalmitoyl-*sn*-glycero-3-phosphocholine (DPPC; Avanti Polar Lipids, Alabaster, AL, USA); egg phosphocholine (Egg-PC; Avanti Polar Lipids, Alabaster, AL, USA); 1,2-dipalmitoyl-*sn*-glycero-3-phosphoethanolamine-N-[methoxy(polyethylene glycol)-2000] (PEG2000 PE); 1,2-dipalmitoyl-*sn*-glycero-3-phospho-(1′-*rac*-glycerol) (DPPG); and cholesterol (Sigma, St Louis, MO, USA) at a molar ratio of 43:28:6:8:15. Xe-liposomes were prepared by the previously described pressurization-freeze method [12]. The rtPA (Alteplase^®^) was obtained from Genentech Inc., San Francisco, CA, USA.

### 2.2. In Vitro rtPA Clot Lytic Activity in the Presence of Xe-Liposomes

To evaluate the effect in vitro, porcine clots were incubated with rtPA either in the presence or absence of Xe-liposomes, and the weight loss of the clots was measured for comparison. The thrombolytic activity of rtPA was assessed using a modified version of the well-established in vitro clot-weight method [25]. Fresh porcine blood samples were collected from non-heparinized Yucatan miniswine, and clots were formed by aliquoting 0.5 mL of blood into 1.5 mL Eppendorf tubes. The tubes were incubated in a 37 °C water bath for 3 h and subsequently stored at 4 °C until further use. The clots were gently blotted, resulting in a weight of 120 mg ± 3.3 mg. Thrombolysis evaluation was conducted by placing the clots in a holder within a 37 °C water bath and treating them with saline, rtPA (80 μg/mL), or Xe-liposomes + rtPA in 1 mL of human plasma containing 1 U/mL human plasminogen for a duration of 30 min. Human plasminogen was used since rtPA clot lysis relies on the activation of plasminogen to form plasmin for fibrinolysis. The activation of human and porcine plasminogen by rtPA has been demonstrated in the presence of the same species or cross-species fibrin [26]. However, the resultant porcine plasmin activity was found to be lower than human plasmin activity by almost 10-fold [26]. Before and after 30 treatments, the clot weights were measured, and the percent clot mass loss was calculated as the difference between the two clot weights divided by the baseline clot weight.

Based on our previous dose-finding study [13], the optimized dose for in vivo intravenous (IV) injection of Xe-liposomes is determined to be 20 mg/kg (6.4 mg lipid/320 mg rat). Another study revealed that the corrected Xe content of Xe-liposomes in rat blood is 23.38 ± 7.36 μL/mg lipid [27]. Considering the total blood volume of 20 mL in a 320 g rat, the estimated Xe concentration in the blood is 7.48 μL/mL. To mimic the in vivo concentration for in vitro experiments, a Xe-liposome concentration of 0.35 mg lipid/mL was employed.

### 2.3. Animal Preparation

All animal experiments were approved by the Animal Welfare Committee at The University of Texas Health Science Center in Houston. Male Sprague Dawley rats (280–320 g, Harlan Laboratories Inc., Indianapolis, IN, USA) were randomly selected as the animal model. Young males were exclusively used in this study considering potential sex differences in the outcome of embolic stroke, which could be influenced by the hormonal cycling of females in response to the Xe treatment. The animals were individually housed in cages in a temperature-controlled room (21–25 °C) with a 12-h light/dark cycle and 45–50% humidity, with free access to food and water. All experimental procedures were conducted in accordance with the approved protocols of the Animal Care and Use Committee of The University of Texas Health Science Center in Houston. Treatments and behavioral tests were conducted in a randomized and blinded manner.

### 2.4. Embolic Middle Cerebral Artery Occlusion (eMCAO) Model

Cerebral ischemia was induced by occluding the right MCA using a clot. The reperfusion was achieved through rtPA-induced clot lysis. On the day of surgery, rats were anesthetized with 5 vol% isoflurane for induction and maintained at 2 vol% throughout the procedure. Rats were placed in a prone position on a rodent surgical monitor (Scintica, London, ON, Canada), which provided surgical warming and vital monitoring of body temperature, ECG, and heart rate. Body temperature was closely monitored and maintained at 37 °C throughout the entire experiment. A hole was carefully drilled at coordinates 1.7 mm posterior and 5.5 mm lateral from the bregma. A laser Doppler flowmetry (LDF) probe (Perimed, Stockholm, Sweden) was positioned onto the right parietal bone to monitor changes in cerebral blood flow (CBF) and confirm the successful induction of cerebral ischemia. A catheter was inserted into the femoral vein for the administration of rtPA. Subsequently, a midline neck incision was performed to expose the right common carotid artery (CCA) under an operating microscope. Ligation of the external carotid artery (ECA) was performed near its distal end, while isolation and separation of the internal carotid artery (ICA) from surrounding tissues were carried out. Incisions were made in the proximal branches of the ECA to facilitate the introduction of a PE-10 catheter. In order to produce homologous blood clots, 200 μL of rat arterial blood was mixed with 10 μL of 0.2 U thrombin and 20 mM CaCl2, followed by withdrawal into a 30 cm PE-50 tubing and incubation at 37 °C for 2 h. The clots were then washed, stored in sterile saline at 4 °C for 22 h, and transferred to distilled water for a 5-min rinsing period. A modified PE-10 catheter was used to draw a clot fragment measuring 12 × 0.35 mm. The catheter, containing the clot, was gently inserted into the right ECA until the tip reached a position just proximal to the origin of the MCA. The clot in the catheter was then injected into the MCA using 80 μL of saline. Regional blood flow was recorded over time using a LDF meter (Perimed, Stockholm, Sweden). Cases with CBF less than 45% and exceeding a 65% reduction in CBF were strictly excluded from the study design.

### 2.5. Experiment I: Xe-Liposome Administration with Early Treatment of rtPA

The aim of this experiment was to evaluate the potential cytoprotective effect of Xe-liposomes in combination with early IV administration of rtPA (10  mg/kg at 2 h) in an eMCAO rat model (Figure 1). Xe-liposomes (6.4  mg/rat) were administered 110 min after stroke onset, followed by the IV administration of rtPA for 1 h (10% bolus and 90% infusion). The study included four experimental groups (*n* = 9 per group): sham control, eMCAO with no treatment control, eMCAO with free rtPA, and eMCAO with Xe-liposomes + free rtPA. Behavioral outcomes were assessed on days 1, 2, and 3 after eMCAO, and infarct size was determined through triphenyltetrazolium chloride (TTC) staining on day 3. In order to assess the potential impact of Xe-liposomes on rtPA thrombolytic activity in vivo, reperfusion of eMCAO was induced by IV administration of rtPA (10 mg/kg) and the alteration in blood flow was measured as an indicator of rtPA clot lytic activity for the two treatment groups (eMCAO/rtPA, eMCAO/Xe-liposomes + rtPA).

### 2.6. Experiment II: Xe-Liposome Administration with Late Treatment of rtPA

This experiment aimed to evaluate the potential synergistic effect of Xe-liposomes in combination with late IV administration of rtPA (10  mg/kg at 4 h) on behavior, histology, and infarct size at days 3 post-eMCAO (Figure 2). Xe-liposomes (6.4  mg/rat) were given at 110 min after stroke onset, while rtPA was administered 4 h after stroke onset. The study consisted of four groups (*n* = 9 per group): sham control, eMCAO with no treatment control, eMCAO with free rtPA, eMCAO with Xe-liposomes + free rtPA. Behavioral outcomes were assessed on days 1, 2, and 3 following eMCAO, while infarct size, neuronal cell death, and hemorrhage analyses were performed on day 3 after stroke onset.

### 2.7. Regional Cerebral Blood Flow Monitoring

Laser Doppler flowmetry is a method of real-time monitoring of blood flow in the area of the brain supplied via the MCA. Monitoring the blood flow change can ensure the accuracy and reproducibility of the occlusion. The MCA regional blood flow was continuously monitored using a straight-needle LDF probe (Perimed, Stockholm, Sweden) connected to a standard laser Doppler monitor (PF5010 LDPM unit, PF5001 main unit, Perimed, Stockholm, Sweden) [28]. A burr hole with 1 mm diameter was carefully drilled through the skull, positioned 2 mm posterior and 2 mm lateral to the bregma, over the ischemic region. A 0.8 mm diameter LDF probe was inserted into the hole. The regional cerebral blood flow was continuously monitored and recorded using a computer connected to the LDF system. The baseline regional cerebral blood flow value was obtained prior to the insertion of the clot. Changes in blood flow during eMCAO and rtPA-induced reperfusion were recorded and compared to the baseline.

### 2.8. Ultrasound-Enhanced Xe-Liposome Administration

A total of 600 μL of Xe-liposomes (10 mg/mL) were IV administered through the tail vein, 5 min prior to IV administration of rtPA. Continuous-wave ultrasound (1 MHz, 100% duty cycle, 0.5 W/cm^2^) was applied over the CCA during Xe-liposome administration for a duration of 5 min to trigger Xe release from Xe-liposomes into the brain as previously described [12,24].

### 2.9. Administration of rtPA

Reperfusion was achieved by IV administration of rtPA (Alteplase, Genentech Inc., San Francisco, CA, USA). A dose of 10 mg/kg of rtPA was IV infused for 60 min. The selected rtPA dose of 10 mg/kg was motivated by the fact that the clinical dose of rtPA (0.9 mg/kg) demonstrated no recanalization in the rat embolic stroke model due to the rat fibrinolytic system being ten times less sensitive than the human [29].

### 2.10. Assessment of Functional Outcomes

All behavioral tests were conducted in a quiet and low-lit room by an observer blinded with respect to the treatment groups. Motor function and neurological outcomes were evaluated on post-surgery days 1, 2, and 3 using beam walking and grid walking tests [12]. Limb placement was assessed by observing the animal’s ability to lift its head and extend its forelimbs toward a table while suspended over the table by its tail (score 0—no response, score 1—sluggish or delayed response, score 2—rapid and fully executed response). Beam walking was evaluated by observing the animal’s ability to traverse a beam (2.5 × 2.5 × 80 cm) while maintaining balance. Response scores were assigned as follows: score 0—no foot slip, score 1—traversed with grasping of the lateral side of the beam, score 2—exhibited difficulty in crawling across the beam but managed to traverse, score 3—required more than 10 s to traverse the beam due to difficulty in walking, score 4—unable to traverse the beam, score 5—unable to move the body or any limb, score 6—unable to stay on the beam for more than 10 s. For the grid-walking test, the animal was placed on a stainless-steel grid floor (40 × 40 × 20 cm with a mesh size of 2 × 2 cm). Furthermore, the occurrences of foot fault errors, characterized by misplacement of a forelimb or hindlimb resulting in falling through the grid, were meticulously documented.

### 2.11. Assessment of Infarct Volume

On day 3 post eMCAO, brain tissue samples were collected. The dissected brains were sectioned coronally into 2 mm thick slices using a Jacobowitz brain slicer. These brain sections were then stained with 2% 2,3,5-triphenyltetrazolium chloride (TTC; Sigma-Aldrich, St. Louis, MO, USA) in phosphate-buffered saline at room temperature for 30 min. TTC is a water-soluble dye that shows a color change in the presence of living cells. In living cells, the mitochondrial enzyme succinate dehydrogenase reduces TTC to form a water-insoluble, light-sensitive compound known as formazan, which imparts a deep red stain to healthy tissue. Conversely, damaged or dead tissue lacks this enzyme and formazan is not produced, resulting in the tissue remaining white. This color alteration is used to identify infarction and penumbra areas in stroke studies.

Following fixation in phosphate-buffered 4% paraformaldehyde solution, the stained sections were subjected to morphometric analysis. Infarct size was quantified using the freely available image processing software, Image-J (Version 1.51), developed by the National Institutes of Health (NIH). Infarct volume was calculated for each slide by multiplying the slide thickness (2 mm), and then summing the volume of all slides to determine the total infarct volume. Subsequently, the normalized infarct volume was determined as a percentage of the whole brain volume.

### 2.12. Double Immunofluorescent Staining of Matrix Metalloproteinase-9 (MMP9)

Immunostaining of brain coronal cryostat sections (10 µm) was performed by utilizing antibodies against MMP9 (diluted 1:500, Abcam, Cambridge, MA, USA) at 4 °C overnight. As a secondary antibody, Alexa Fluor 488 Goat anti-rabbit IgG was used (diluted 1:4000, Thermo Scientific, Rockford, IL, USA). In order to counterstain the cerebral artery, the sections were incubated with α-smooth muscle (α-SM) actin antibody against rat microvascular smooth muscle antigens (diluted 1:500, Sigma-Aldrich, St. Louis, MO, USA). Alexa Fluor 555 Goat anti-mouse IgG (diluted 1:1000, Life technologies, Grand Island, NY, USA) was used as the secondary antibody. The fluorescently labeled slides were then analyzed under a fluorescence microscope (Nikon, Ti-U, Dusseldorf, Germany).

### 2.13. In Situ Labeling of DNA Fragmentation

To detect DNA fragmentation in degenerating neurons, coronal brain sections (5 μm) were obtained using a cryotome and subjected to the terminal deoxynucleotidyl transferase dUTP nick end labeling (TUNEL) assay (Roche Diagnostics GmbH, Mannheim, Germany). In brief, the sections were fixed in 4% paraformaldehyde for 20 min at room temperature and then permeabilized by incubating with 0.1% Triton X-100 and 0.1% sodium citrate for 2 min at 4 °C. For TUNEL assay, each slide was incubated with 50 μL of TUNEL reaction mixture at 37 °C for 1 h. Additionally, the slides were stained with 4′,6-diamidino-2-phenylindole (DAPI, Invitrogen, Eugene, OR, USA) to visualize the nuclei. Images were captured using a Nikon ECLIPSE Ti fluorescence microscope equipped with a CoolSNAP photomatrics camera. TUNEL-positive cells were photographed at 100× magnification from three randomly selected fields close to the infarct border. The number of dead cells was counted using NIS-Elements BR 3.2 diagnostic software (Nikon Instruments Inc., Melville, NY, USA).

### 2.14. Statistical Analysis

In this preclinical animal study, due to the small sample sizes within the experimental groups, nonparametric statistical analysis methods were used to assess differences between the groups. Pairwise comparisons of spontaneous Xe release from Xe-ELIP in PBS or HSP and release with or without ultrasound were performed by the Wilcoxon rank sum test for two groups. For comparisons of multiple groups, the Kruskal–Wallis analysis of variance of ranks and median test was used to assess if there were global differences between the groups. This was followed by post hoc multiple comparisons of mean ranks for all groups by computing normal z-values for each comparison followed by probabilities adjusted for the number of comparisons for two-sided tests of significance. Data were plotted as means and standard deviation. Neurological outcomes between the treatment groups were reported as medians and quartiles. Statistica (Version 9, StatSoft Inc., Tulsa, OK, USA) software was utilized for the statistical analyses. A *p* < 0.05 was considered significant.

## 3. Results

All sample size determinations were made at α = 0.05. A sample size of more than seven per group was determined to ensure a statistical power of at least 90% to detect the expected difference. Survival, change of regional blood flow in response to occlusion and treatment, and gross hemorrhage were continuously monitored and recorded for both studies of Xe-liposomes in combination with early rtPA administration (Table 1) and late rtPA administration (Table 2). Total animals refer to the animals that have successfully undergone either sham surgery or eMCAO surgery with an eligible occlusion rate. Survival refers to the animals that have successfully survived longer than day 3 after the surgery.

In the early rtPA treatment group, 39 animals successfully underwent eMCAO surgery with eligible occlusion rates (Table 1). A total of 17 animals were included in the eMCAO group without any treatment. Out of these, 10 animals died following eMCAO, while 7 animals survived until day 3 after surgery without any treatment. Subsequently, 11 eMCAO animals received treatment with rtPA at 2 h after stroke, among which 3 animals did not survive while 8 animals successfully survived until day 3 after surgery. Another set of 11 eMCAO animals were treated with both Xe-liposomes and rtPA. In this combined treatment group, 3 animals did not survive, similar to the rtPA-only group, and 8 animals successfully survived until day 3 after surgery.

In the late rtPA treatment group, 40 animals successfully underwent eMCAO surgery with eligible occlusion rates (Table 2). A total of 17 animals were included in the eMCAO group without any treatment. Out of these, 10 animals died following eMCAO, while 7 animals survived until day 3 after surgery without any treatment. Another set of 12 eMCAO animals received treatment with rtPA at 4 h after stroke. Among them, 4 animals did not survive, while 8 animals survived until day 3 after surgery. Lastly, a set of 11 eMCAO animals were treated with both Xe-liposomes and rtPA. Out of these animals, 3 did not survive, while 8 animals successfully survived until day 3 after surgery.

### 3.1. No Inhibition of rtPA Thrombolytic Efficiency in the Presence of Xe-Liposomes In Vitro

In order to investigate the thrombolytic efficiency of rtPA in the presence of Xe-liposomes, an in vitro clot lysis assay was conducted (Figure 3). Free rtPA treatment revealed a percentage clot weight loss of 34 ± 3.0%. The combination treatment of Xe-liposomes with rtPA achieved an effective clot lysis of 33 ± 8.8%, comparable to that of free rtPA alone. These findings demonstrate the comparable thrombolytic potential of Xe-liposomes with rtPA. The next steps involve assessing the safety and efficacy of this combination in both in vivo and clinical-related settings.

### 3.2. Xe-Liposomes in Combination with Early rtPA Administration without Interfering with rtPA-Induced Reperfusion

The therapeutic efficacy of Xe-liposomes in conjunction with IV rtPA was assessed in a rat eMCAO model. Male Sprague Dawley rats were subjected to eMCAO by injecting a 12 mm long blood clot into the MCA to induce ischemia. Continuous monitoring of MCA regional blood flow was performed to verify the success of occlusion and to assess the effects of rtPA-induced reperfusion. Rats with eMCAO resulting in regional blood flow reduction within the range of 50–60% were selected for inclusion in the experimental group. In order to determine if Xe-liposomes interfere rtPA activity, regional blood flow was monitored before and after the rtPA treatment with the presence and absence of Xe-liposomes. The MCA regional blood flow change compared to the baseline for eMCAO after free rtPA treatment was 21 ± 7.9%, while the eMCAO after Xe-liposomes combined with rtPA treatment was 23 ± 5.5% (*p* = 0.96 vs. free rtPA; Table 1).

The therapeutic effect of Xe-liposomes in combination with early IV rtPA (10 mg/kg) was evaluated (Figure 1). The untreated thrombotic stroke control group demonstrated the largest infarcts, accounting for approximately 16.5 ± 6.0% of the whole brain (Figure 4a). Treatment with IV rtPA (10 mg/kg at 2 h) reduced the infarct size to 5.6 ± 3.0% (*p* < 0.05 vs. no treatment). When Xe-liposomes were administered in combination with rtPA, a further reduction in infarct size to 2.4 ± 0.9% (*p* < 0.05 vs. rtPA treatment) was observed. The mortality rates for embolic strokes with no treatment, treated with rtPA alone, and treated with the Xe-liposome + rtPA combination were 59%, 27%, and 27%, respectively (Table 1). There was no difference in core body temperature between the groups during eMCAO and the initial hours of reperfusion. These findings collectively demonstrate the effective cerebroprotection by the combined treatment of Xe-liposomes and rtPA in ischemic stroke.

The impact of combining Xe-liposomes with early rtPA treatment on the neurological disability of animals with ischemia due to eMCAO occlusion was investigated (Figure 4b–d). Untreated animals exhibited substantial locomotor impairments on day 1, with little improvement on days 2 and 3. The group treated with rtPA only demonstrated improved performance in all assessments from day 2, particularly reaching near-normal performance level on day 3 in the limb placement test (*p* < 0.05 vs. no treatment group). Animals subjected to Xe-liposomes along with rtPA treatment demonstrated further enhanced neurological improvement across all tests, surpassing both the untreated control and rtPA-only groups.

### 3.3. Xe-Liposomes in Combination with Late rtPA Administration

In the late rtPA treatment group, rtPA was administrated at 4 h after stroke onset (Figure 2). The untreated thrombotic stroke resulted in the largest infarct size, occupying a volume of 20.7 ± 6.1% of the brain (Figure 5a). However, rtPA treatment reduced the infarct size to 7.6 ± 2.7% of the brain volume (*p* < 0.001 vs. no treatment). Xe-liposomes in combination with the rtPA treatment further reduced the infarct size to 2.0 ± 1.4% of the brain volume (*p* = 0.032 vs. rtPA treatment).

Behavioral deficits were found to correlate with infarct volume outcomes (Figure 5b–d). Animals without any treatment showed major deficits in locomotor performance on day 1 with minimal improvement on subsequent days. It was noteworthy that the animals treated solely with rtPA failed to yield substantial improvements in any behavioral test. However, the animals treated with Xe-liposomes in combination with rtPA demonstrated markedly improved neurological performance across all tests (Figure 5b–d).

### 3.4. Evaluation of Cell Death

The evaluation of cell death at 72 h post-ischemic stroke demonstrated a decrease in the number of TUNEL-labeled cells (Figure 6a). The number of TUNEL-positive cells (cells/mm^2^) was quantified to evaluate cell death. In the affected tissue around the ischemic core, the Xe-liposomes + rtPA treatment group exhibited 155 ± 81 cells/mm^2^ with TUNEL positivity, significantly lower than the rtPA-only group (324 ± 37 cells/mm^2^; *p* = 0.015), the untreated eMCAO group (593 ± 46 cells/mm^2^; *p* < 0.001), and the sham control group (24 ± 10 cells/mm^2^; *p* = 0.017) (Figure 6b).

### 3.5. MMP9 Immunoreactivity

Given the well-known association between the MMP9 pathway and BBB integrity, as well as its role in hemorrhagic transformation following cerebral ischemia [30,31], the expression of MMP9 and its colocalization with cerebral microvessels were investigated. The treatment group with Xe-liposomes and late rtPA administration exhibited a noticeable reduction in MMP9 immunoreactivity compared to the rtPA-only group (Figure 7). Furthermore, MMP9 immunoreactivity was predominantly observed within the microvessel walls.

## 4. Discussion

The efficacy of current treatments using cytoprotective drugs for acute stroke remains limited. Many studies have focused solely on the administration of these drugs without considering their integration with IV rtPA treatment. At present, the primary strategy in treating acute ischemic stroke involves vascular reperfusion facilitated by thrombolysis (rtPA) and endovascular thrombectomy. The aim of cytoprotection is to forestall or mitigate brain cell death and minimize infarct size resulting from stroke. To achieve this goal, it is crucial to ensure that neuroprotective agents are compatible with rtPA treatment.

In our earlier investigations, we presented compelling evidence for the efficacy of Xe-liposomes in preserving brain tissue from ischemic injury and maintaining neurological function in a transient MCAO rat model. To further translate our findings to potential clinical application, we investigated the compatibility of Xe-liposomes with IV rtPA treatment and assessed whether this combination could offer comparable benefits in the treatment of stroke.

In the present study, we examined the effects of Xe-liposome cerebroprotection combined with rtPA on rtPA thrombolytic activity. Our findings demonstrated that Xe-liposomes did not inhibit rtPA thrombolytic activity in vitro or in vivo. The transformation of plasminogen to plasmin, a fibrinolytic enzyme crucial for clot dissolution, is catalyzed by rtPA. In a previous in vitro clot lysis study, human plasminogen (1 μg/mL) was used instead of porcine plasminogen due to the markedly reduced activation of porcine plasminogen by rtPA compared to that of human plasminogen [26]. The typical concentration range of plasminogen in human blood falls within 0.75–1.60 μ/mL [32].

Remarkably, animals treated with the combination of Xe-liposomes and rtPA exhibited superior cognitive function compared to those receiving rtPA alone. An intriguing trend was also observed, revealing a decrease in rtPA-induced hemorrhagic transformation during late rtPA treatment when combined with Xe-liposomes. We postulate that this effect could be attributed to the stabilization of the BBB by rtPA. These findings demonstrate the potential of Xe-liposomes as a beneficial cytoprotective agent when combined with rtPA for the treatment of ischemic stroke.

Prior studies have provided diverse insights into the impact of Xe on rtPA activity. David et al. investigated the influence of different Xe concentrations on the thrombolytic efficiency of rtPA [33]. Their findings revealed an inhibitory effect of Xe on the fibrinolytic activity of rtPA at higher concentrations (50%), affecting its ability to dissolve blood clots both in vitro and in vivo. In response to concerns regarding reduced protein activity due to direct sparging of protein solutions with noble gases [34,35,36], Cahill et al. [37] employed an indirect gas delivery method. Both computational modeling and experimental data demonstrated minimal interaction of Xe (1–3 mM) with enzymes. This implies that the inhibitory effects of Xe on enzymes, as reported in previous studies, may have been attributed to protein damage caused by directly bubbling gases into protein solutions. To further elucidate this dynamic interaction, we performed a detailed evaluation of the effect of Xe-liposomes on rtPA thrombolytic activity both in vitro and in vivo. Our findings suggest that Xe-liposomes do not impede the thrombolytic action of rtPA in vivo. Furthermore, the in vitro experiments demonstrated no significant difference between rtPA administered alone and rtPA combined with Xe-liposomes.

Combining Xe-liposome administration with both early and late rtPA therapy resulted in improved neurological outcomes. The proposed mechanisms underlying the cerebroprotective effects of Xe-liposomes involve pleiotropic pathways including neuroprotection, anti-apoptosis, anti-inflammation, and protection against ischemic-reperfusion injury [38,39]. Recently, proteomic analysis has revealed that Xe interacts with approximately 90 proteins [38], several of which play crucial roles in ATP-dependent processes and endothelial stability [38,40].

Despite the efficacy of rtPA in clot lysis, its administration at the later stages of stroke is associated with an increased risk of hemorrhagic transformation. Consequently, there is a growing need to develop novel therapies aimed at improving stroke outcomes by stabilizing brain tissue. In the present study, the effect of Xe-liposomes on reducing infarct size and behavioral impairments was demonstrated. Moreover, Xe-liposomes showed potential for modulating MMP9 and stabilizing the BBB. A parallel study focused on validating the detailed mechanism underlying the effect of Xe on the BBB and MMP9 is currently underway, and its results will be published independently. Nonetheless, the crucial aspect is that Xe does not exacerbate rtPA-induced hemorrhagic transformation. This valuable safety and efficacy data will help future research aimed at validating the therapeutic potential of Xe-liposomes in larger animal models and clinical trials.

Several limitations need to be acknowledged in the present study. Infarct size assessment on day 3 without factoring in edematous effects might lead to an underestimation of the full extent of ischemic injury. Edema, comprising cytotoxic and vasogenic components, is a common reaction to cerebral infarction and leads to brain tissue swelling. This swelling can extend beyond the boundaries of the initial infarcted area due to vasogenic edema. The most substantial increment in brain tissue swelling is noted at the 24-h mark post-injury [41]. Failing to account for edema during the evaluation can result in a smaller measurement of the infarct size, as the edematous tissue would not be taken into consideration. This underestimation could lead to an inadequate understanding of the actual impact of the therapeutics. Moreover, insufficient conclusive evidence currently exists to establish a definitive correlation between Xe and its impact on MMP9, as our investigation solely focused on alterations in the immunoreactivity of MMP9 within microvessels. To establish a more comprehensive understanding, it is imperative for further rigorous research to quantify the extent of MMP9 variations. Such an approach should encompass a comprehensive evaluation of the consequences following the disruption of the BBB integration.

## 5. Conclusions

Our study built upon previous research that demonstrated that the co-administration of Xe-liposomes and rtPA effectively reduced infarct volume and improved neurological function in thrombotic stroke animal models. We extensively investigated the compatibility of Xe-liposomes with rtPA in a clinically relevant embolic stroke model and confirmed no interference with the thrombolytic activity of rtPA both in vitro and in vivo. The combination treatment of rtPA with Xe-liposomes demonstrated superior outcomes, maintaining the thrombolytic activity of rtPA while minimizing the risk of hemorrhagic transformation. Our study suggests that the combined therapy of Xe-liposomes and rtPA may provide cerebroprotection and improve neurological function recovery in cases of acute stroke. The implications of these findings underscore the need for further investigation to determine the feasibility of translating this approach from pre-clinical studies to human clinical trials. In a clinical setting, Xe-liposomes could be pre-administered, co-administered, or post-administered with rtPA to maximize treatment efficacy.

## Figures and Tables

**Figure 1 biomolecules-13-01256-f001:**
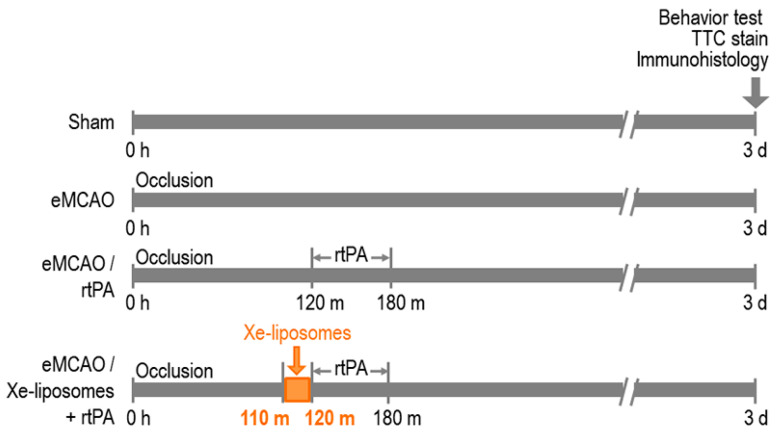
Experimental design of Xe-liposome administration with early treatment of rtPA. Four experimental groups were: sham control, eMCAO with no treatment control, eMCAO with free rtPA, and eMCAO with Xe-liposomes + free rtPA. Xe-liposomes (6.4  mg/rat) were administered 110 min after stroke onset, followed by IV administration of rtPA (10% bolus and 90% infusion) for 60 min to induce reperfusion. Behavioral outcomes were assessed on days 1, 2, and 3 after eMCAO, and infarct size was determined on day 3.

**Figure 2 biomolecules-13-01256-f002:**
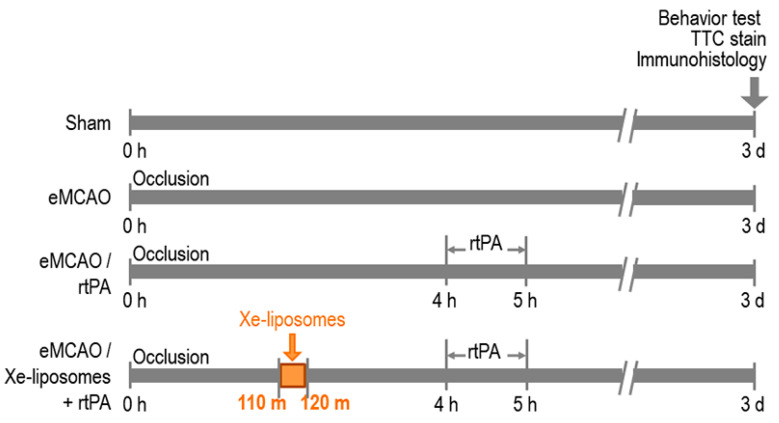
Experimental design of Xe-liposome administration with late treatment of rtPA. Four experimental groups were: sham control, eMCAO with no treatment control, eMCAO with free rtPA, and eMCAO with Xe-liposomes + free rtPA. Xe-liposomes (6.4  mg/rat IV) were given at 110 min after stroke onset, while rtPA was administrated at 4 h after stroke. Behavioral outcomes were assessed on days 1, 2, and 3 after eMCAO. Immunohistology staining and infarct size were analyzed on day 3.

**Figure 3 biomolecules-13-01256-f003:**
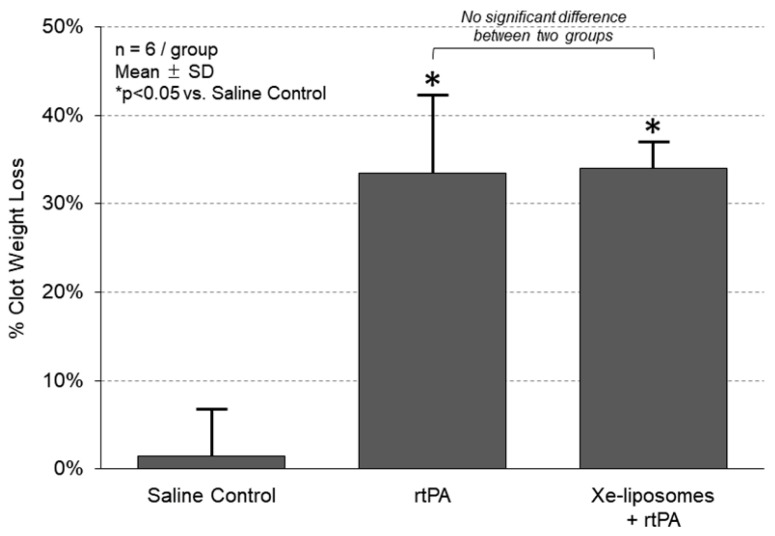
Thrombolytic efficiency of rtPA in the presence of Xe-liposomes in vitro. The thrombolytic activity of rtPA in the presence and absence of Xe-liposomes was evaluated using a porcine clot lysis assay. Percentage clot weight loss was measured as an indicator of thrombolytic efficiency.

**Figure 4 biomolecules-13-01256-f004:**
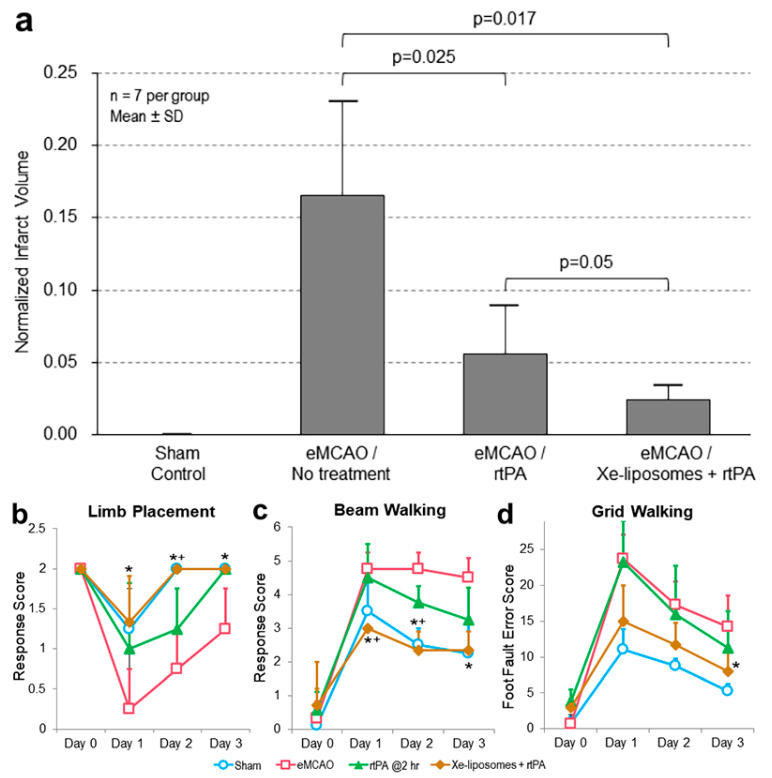
Therapeutic efficacy of Xe-liposomes in combination with early rtPA thrombolysis. (**a**) Quantitative comparison of infarct size among the groups, and neurological assessments of (**b**) limb placement, (**c**) beam walking, and (**d**) grid walking. *n* = 7/group, Mean ± SD; * *p* < 0.05 vs. eMCAO, ^+^ *p* < 0.05 vs. rtPA.

**Figure 5 biomolecules-13-01256-f005:**
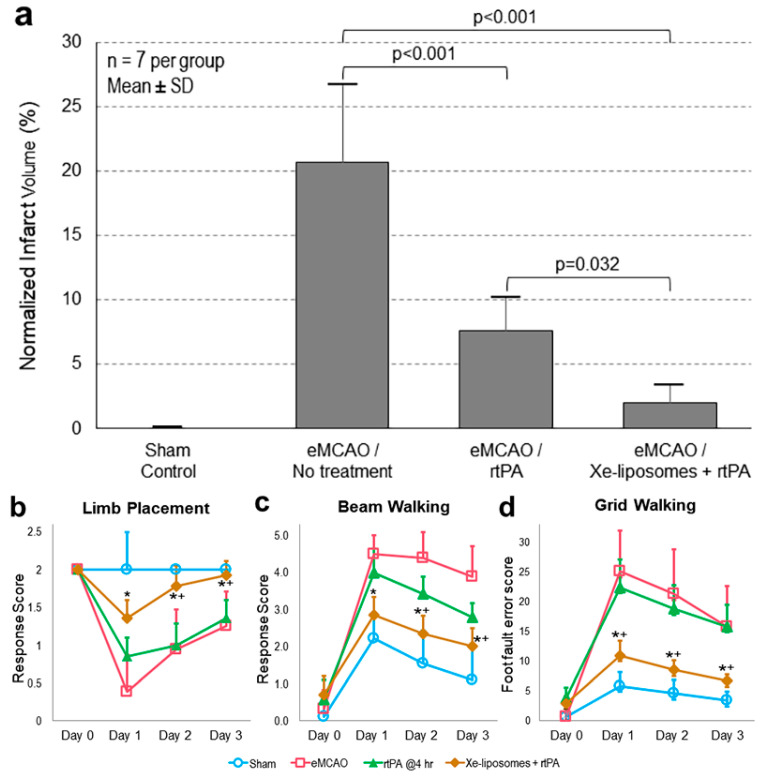
Therapeutic efficacy of Xe-liposomes in combination with late rtPA thrombolysis. (**a**) Quantitative comparison of infarct size among the groups, and neurological assessments of (**b**) limb placement, (**c**) beam walking, and (**d**) grid walking. *n* = 7/group, Mean ± SD; * *p* < 0.05 vs. eMCAO, ^+^ *p* < 0.05 vs. rtPA.

**Figure 6 biomolecules-13-01256-f006:**
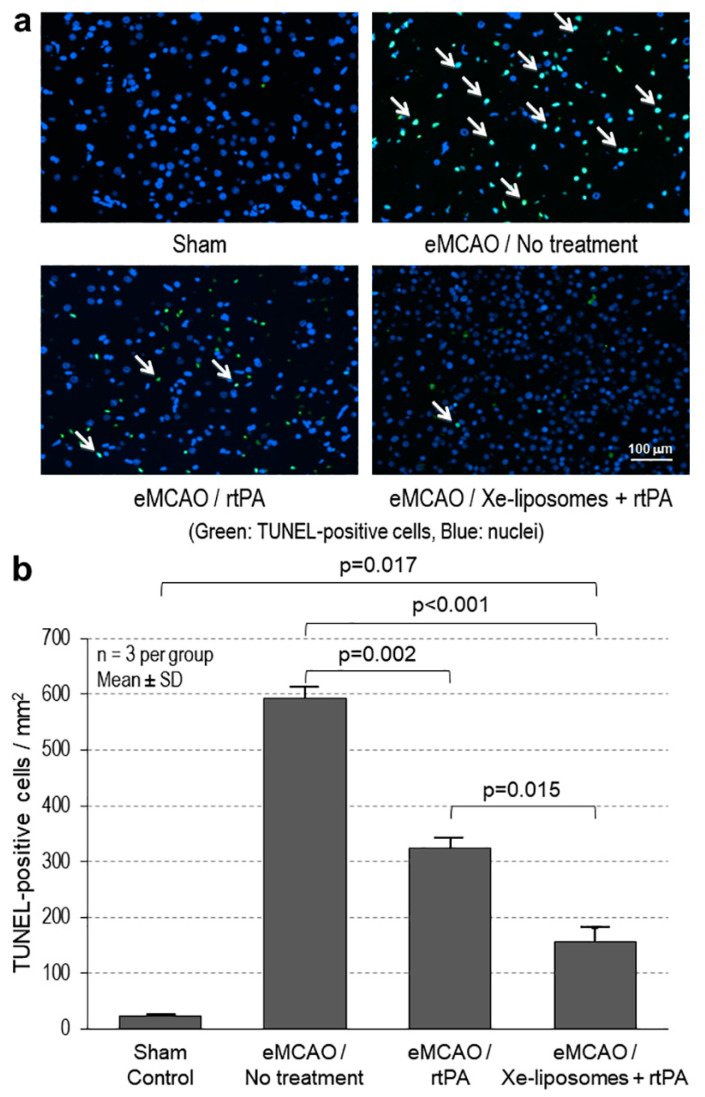
Reduced cell death after ischemic stroke with Xe-liposome treatment. (**a**) Representative images of the TUNEL-labeled cells in the affected tissue region around the ischemic core of the brain sections from the sham, eMCAO with no treatment, eMCAO with rtPA treatment alone, and eMCAO with Xe-liposomes + rtPA treatment groups. Arrows indicate the examples of the TUNEL positive cells. (**b**) Quantitative evaluation of the TUNEL-labeled cells.

**Figure 7 biomolecules-13-01256-f007:**
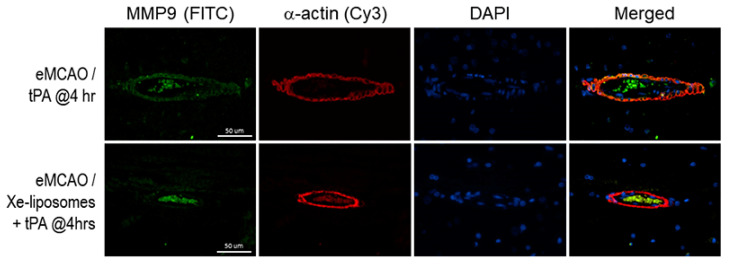
Colocalization of MMP9 immunoreactivity on the cerebral microvascular wall. In the late rtPA treatment group, MMP9 immunoreactivity was observed extensively around the cerebral microvascular wall, co-localizing with α-actin. However, administration of Xe-liposomes along with rtPA showed large reduction in MMP9 immunoreactivity on the microvascular wall.

**Table 1 biomolecules-13-01256-t001:** MCA regional blood flow change and survival rate of the animal groups for Xe-liposomes combined with early rtPA administration.

Group	Total Animals	Survival	Occlusion Rate	Reperfusion Rate Post-IV rtPA
Sham	9	9	0	n/a
eMCAO/ No treatment	17	7	56 ± 9%	0
eMCAO/ rtPA at 2 h	11	8	50 ± 8%	21 ± 7.9%
eMCAO/ Xe-liposomes + rtPA at 2 h	11	8	53 ± 6%	23 ± 5.5%

**Table 2 biomolecules-13-01256-t002:** MCA regional blood flow change and survival rate of the animal groups for Xe-liposomes combined with late rtPA administration.

Group	Total Animals	Survival	Death	Occlusion Rate	Hemorrhage	Hemorrhagic Rate
Sham	9	9	0	0%	0	0%
eMCAO/ No treatment	17	7	10	56 ± 9%	1	13%
eMCAO/ rtPA at 4 h	12	7	5	50 ± 14%	7	58%
eMCAO/ Xe-liposomes + rtPA at 4 h	11	7	4	46 ± 17%	2	20%

## Data Availability

The data presented in this study are available on request from the corresponding author. All data generated or analyzed during this study are included in the manuscript.

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
