# Peer review of "Enhanced Cerebroprotection of Xenon-Loaded Liposomes in Combination with rtPA Thrombolysis for Embolic Ischemic Stroke"

_biomolecules, 2023, doi:10.3390/biom13081256_

Round 1

Reviewer 1 Report

In the paper entitled:” Enhanced Cerebroprotection of Xenon-loaded Liposomes in Combination with rtPA Thrombolysis for Embolic Ischemic Stroke”, the authors describe the combination of Xe-liposomes with rtPA can provide an improved therapeutic efficacy when compared with rtPA treatment alone.

Several questions must be addressed before considered to be published in Biomolecules:

1.         Table 1 and table 2 are confusing. In table 1 and table 2, how occlusion rates and reperfusion rates were calculated? The denominator (i.e., total animals or survived ones) must be defined to avoid confusion.

2.         Following previous question and take Table 1 as example: in “eMCAO /No treatment” group, the survival rate was 41% while the occlusion rate was 56 ± 9%. The results indicate that the death could be resulted from the surgery and from the induced eMCAO, which makes the reader hard to follow. The successful rate in generating eMCAO model and the death resulted from the eMCAO should be separated.

3.         The author attributed the potential in reducing delayed rtPA-induced brain hemorrhagic transformation to Xe-liposomes’ ability to modulating MMP9 levels and stabilizing the BBB. Therefore, in MMP9 study, a Xe-liposome alone group is highly recommended to support the statement (mechanism as addressed is not needed in current manuscript).

4.         There were flaws in statistical analysis, which needs to be improved:

a. The *p > 0.05 in Figure 3 is confusing. What the authors would like to express is the combination of Xe-liposomes with rtPA presented comparable/similar therapeutic effect to rtPA alone; then, why do not just label no significant difference between two groups?
b. Why did the authors use mean
± SEM instead of mean ± SD, which is considered to be more accurate especially when the number in animal study is low?
c. Symbols such as * and + have to be described in Figure 4-5. Also, the authors should also describe the post-hoc test they use in figure legend or statistical analysis section. 

Reviewer 2 Report

Xenon (Xe) exhibits promising potential as a stroke treatment for brain tissue protection without notable side effects. The authors developed Xe-loaded liposomes for targeted delivery and assessed their efficacy in combination with recombinant tissue plasminogen activator (rtPA) in an embolic stroke model. Through in-vivo experiments, they demonstrated successful clot lysis and reduced cell death when utilizing Xe-liposomes in conjunction with rtPA. Additionally, the combination therapy proved effective in mitigating rtPA-induced hemorrhage by decreasing MMP9 . This study highlights the enhanced therapeutic prospects of employing Xe-liposomes alongside rtPA for treating strokes. The research conducted reached a commendable level of methodological rigor. However, further clarification of the study's details is necessary before publication.

The statement "Ischemic strokes account for approximately 87% of all stroke cases" should be accompanied by a literary reference.

Regarding the statement: "Upon the occurrence of an ischemic stroke, interruption of the blood supply leads to critical consequences such as the initiation of inflammatory responses, activation of cell death pathways, and the potential for salvaging additional brain tissue," supporting references specific to each consequence should be provided.

The article does not specify the method utilized to normalize the concentration of human plasminogen in human plasma containing it. Additional information or references are required to address this question adequately.

The manuscript does not provide a clear explanation as to why clot weights were evaluated after 30 minutes of treatment despite the clot being incubated with rtPA for 2 hours.

Could you please clarify whether the rats were pre-trained in the beam-walking test?

There is no information provided in the article regarding whether normalization was performed taking into account the edema of the damaged hemisphere when calculating the volume of brain damage.

Behavioral test scores, such as those mentioned in the article, are typically considered non-parametric data and are commonly presented as a median and interquartile range .

The titles of tables 1 and 2 need clarification to indicate which experimental scheme they refer to. Clear and specific table titles are important for readers to understand the context of the data presented.

The captions in figures 4 and 5 need clarification by incorporating detailed statistical analyses and explaining which groups are being compared. This would enhance the interpretation of the data presented.

The authors provide images of brain sections stained with TTC. Whether these images are representative and correspond to the average values for each group presented in the figures. The reviewer considers it useful to provide images for all animals in additional files.

It should be noted that the authors of the manuscript do not provide detailed descriptions of the observed changes in behavioral studies, instead referring only to the figures. To enhance the comprehensiveness of their findings, it is recommended for the authors to provide more in-depth descriptions of the results.

In section "3.3. Xe-liposomes in combination with late rtPA administration with reduced hemorrhagic transformation," there is no description of hemorrhagic transformation changes. Furthermore, there is no methodological description of measuring hemorrhagic transformation. To address this issue, a detailed explanation of the methods used to evaluate hemorrhagic transformation and statistical analysis should be provided.

How did the authors determine the penumbra zone by cell count using the TUNEL method? It is well known that the penumbra zone is present for a short time in the brain, as the authors suggest, can the penumbra zone remain 72 hours after ischemia/reperfusion simulation?

What do the authors mean by the term “inhibition” in the heading “3.5. MMP9 inhibition by Xe-liposomes? The authors describe only morphological changes in micrographs. It is necessary to quantify the resulting changes in MMP9. But this study does not answer the question about the activity of MM9. It is necessary to conduct additional studies using the “Gelatin zymography” method. Also, a consequence of the activity of MMPs 9 and 3 is the disruption of the BBB, the integrity of which must be proved in this study.

The following conclusion “Moreover, Xe-liposomes showed promising potential in reducing delayed rtPA-induced brain hemorrhagic transformation, potentially by modulating MMP9 levels and stabilizing the BBB. ” is not justified by experimental data that would be subject to statistical analysis.

Readers would benefit from reviewing the full range of known neuroprotective mechanisms of xenon, including the regulation of gene expression (PMID: 34356124). 

Round 2

Reviewer 2 Report

The authors have made substantial revisions to the manuscript, addressing all the questions raised by the reviewer. Notably, they have added a section highlighting the limitations of their study. This inclusion demonstrates that the authors have taken the reviewer's feedback seriously and have provided a more comprehensive and balanced discussion of their findings. With these revisions in place, the revised manuscript is now suitable for publication.